# DDT Resistance in *Anopheles pharoensis* from Northern Cameroon Associated with High Cuticular Hydrocarbon Production

**DOI:** 10.3390/genes13101723

**Published:** 2022-09-25

**Authors:** Nelly Armanda Kala-Chouakeu, Edmond Kopya, Vasileia Balabanidou, Borel Tchamen Djiappi, Kyriaki Maria Papapostolou, Timoléon Tchuinkam, Christophe Antonio-Nkondjio

**Affiliations:** 1Vector-Borne Diseases Laboratory of the Applied Biology and Ecology Research Unit (VBID-URBEA), Department of Animal Biology, Faculty of Science, University of Dschang, Dschang P.O. Box 067, Cameroon; 2Laboratoire de Recherche sur le Paludisme, Organisation de Coordination Pour la lutte contre les Endémies en Afrique Centrale (OCEAC), Yaoundé P.O. Box 288, Cameroon; 3Faculty of Sciences, University of Yaoundé I, Yaoundé P.O. Box 337, Cameroon; 4Institute of Molecular Biology and Biotechnology, Foundation for Research and Technology Hellas, 71110 Heraklion, Greece

**Keywords:** DDT resistance, *Anopheles pharoensis*, cuticular hydrocarbon, malaria, Cameroon

## Abstract

Despite the contribution of secondary vectors to malaria transmission, there is still not enough information on their susceptibility status to insecticides. The present study assesses the resistance profile of *Anopheles pharoensis* to DDT. WHO tube tests were used to screen mosquito populations collected from the far-north region of Cameroon for susceptibility to 4% DDT. High DDT resistance in *An. pharoensis* populations from Maga, Simatou and Yangah with mortality rates ranging from 62.79% to 80% was recorded. Direct sequencing (Sanger) of the VGSC gene was undertaken to search for *kdr* L1014F/S mutations. However, no *kdr* allele was detected in the resistant samples. We then looked for cuticle alterations and CHC identification and quantitation were undertaken using GC-MS and GC-FID. High production of cuticular hydrocarbon was recorded in the populations of Yangah and Simatou, with 2420.9 ± 265 and 2372.5 ± 225 ng CHCs/mg dry weight, respectively. The present findings are the first ever describing the development of cuticle resistance in *An. pharoensis*. The data suggest the need to expand surveillance activities on other vector species.

## 1. Introduction

Cameroon is among the eleven countries most affected by malaria in the world [1]. Alongside the rapid expansion of insecticide resistance in malaria vectors, the continuous burden of malaria in Cameroon also results from the diversity of vector species taking part in transmission [2,3]. Over 17 species have been reported as major or secondary malaria vectors in the country [2,3]. In dry savannah areas, apart from the major vectors *An. gambiae* s.l. and *An. funestus*, species such as *An. pharoensis* are frequently found to be infected with Plasmodium falciparum and thus contribute to malaria transmission [2,4]. *An. pharoensis* is a species which has been reported in Cameroon since the 1950s, mainly distributed in the northern part of the country, and is considered to display an opportunistic feeding behaviour, feeding predominantly on domestic animals [2]. *An. pharoensis* larvae breed in large vegetated swamps along the shores of lakes in the grass zone. The species is also very common in rice fields, along the edges of streams and temporary flooded areas [5]. *An. pharoensis* is considered to be distributed in many places across Africa. Its distribution extends from Mauritania to Sudan in the north and to South Africa in the south. Many recent studies have documented its presence in countries such as Ethiopia, Nigeria, DRC, Tanzania, Uganda, Cameroon, Benin, Chad, Burundi, Senegal [2,6,7,8,9]. In most places, the species is abundant towards the end of the rainy season; it can also be recorded in high densities during the dry season when flooded areas persist [5]. *An. pharoensis* is capable of remarkably long migratory flights [10]. The species bites both indoors and outdoors and enters houses at night, but few rest indoors. Studies conducted in Ethiopia indicated increasing resistance level in this species to different compounds including DDT and pyrethroids [11].

Despite the increasing number of studies reporting the rapid expansion of insecticide resistance in both *An. gambiae* and *An. funestus* in Cameroon, there is still not enough data on the susceptibility status and profile of resistance mechanisms in other malaria vectors to insecticide commonly used in public health. Studies conducted recently indicated the emergence of insecticide resistance in species such as *An. nili*, *An. paludis* and *An. rufipes* [6,12]. In the frame of the present study, we investigated the resistance profile of *An. pharoensis* populations from northern Cameroon to DDT.

## 2. Materials and Methods

### 2.1. Description of the Study Sites

Mosquito collections took place in the villages of Simatou (10°52′24″ N, 14°58′48″ E), Yangah (14°44′23″ N, 14°58′35″ E) and Maga (10°50′36″ N, 14°56′23″ E), all located in the far-north region of Cameroon (Figure 1). The region belongs to the Sahelian domain, characterized by a long dry season running from October to May with only four months of rains (June to September), with vegetation consisting of shrubby savannah. These villages are situated close to a large dam (Maga Dam) and the Logone river. The population mainly practice farming, fishing and cattle rearing for a living. The main crops cultivated are rice, cotton, maize and groundnuts. Although the area is prone to seasonal malaria transmission, the presence of a dam has created suitable habitats for the development of mosquitoes all year round.

### 2.2. Mosquito Collection, Rearing and Conservation

Field sampling of anopheline larvae and processing was conducted during the rainy season from August to September 2021 in Maga, Simatou and Yangah. Larval collections were undertaken in typical habitats for *An. pharoensis* larvae, which are permanent or semi-permanent water collections with vegetation. In each study site, collected samples were pooled per site and reared to the adult stage. Larvae were fed with TetraMin^®^ fish food until pupae. Pupae were collected in cardboard cups and placed in netting cages for adult emergence. After emergence, adults were offered sugar solution until processing. A subset of 30–40 unexposed, non-blood fed, 3–5 days-old female *An. pharoensis* were air-dried and sent to the Laboratory of Molecular Entomology in IMBB-FORTH for analysis (CHC quantification). Another subset of about 25–100 mosquitoes per population was used for insecticide bioassays; survivors after exposure to insecticide were preserved in 70% alcohol for molecular analysis.

### 2.3. Insecticide Bioassay

Adult female *An. pharoensis* aged 3–5 days reared from larval collections in different sites were tested against DDT 4% following WHO guidelines [13]. *An. pharoensis* females were placed in batches of 20 to 25 mosquitoes per tube and left for observation for one hour. After this period, mosquitoes were transferred in tubes with insecticide-impregnated papers and exposed for 1 h. The susceptible laboratory strain (*An. gambiae* (Kisumu strain)) was used as control to assess the quality of the impregnated papers. The number of mosquitoes knocked down by the insecticide was recorded after 1 h of exposure; then, mosquitoes were transferred to holding tubes and fed for 24 h before scoring the mortalities. Mosquitoes were considered resistant when the mortality rate was <90% and susceptible when the mortality rate was ≥98%, and resistance status needed further checking when mortality rate was <98% and ˃90% [14].

### 2.4. Genomic DNA Extraction from Individual Mosquitoes

Genomic DNA was extracted from 72 specimens for molecular species identification using the DNazol protocol, according to the manufacturer’s instructions (Molecular Research Center Inc., Cincinnati, OH, USA). The quantity and purity of DNA were assessed spectrophotometrically via Nanodrop measurements. The quality of DNA was assessed by 1.0% *w*/*v* agarose gel electrophoresis.

### 2.5. Total Nucleic Acid (NA) Extraction from Mosquito Pools

Total NAs were extracted from mosquito pools using a magnetic-bead-based approach with the MagSi kit (MagnaMedics Diagnostics GmbH, Aachem, Germany) for direct sequencing (Sanger) of the VGSC gene. The quantity and purity of DNA and total RNA were assessed spectrophotometrically via Nanodrop measurements. The quality of RNA was assessed by 1.0% *w*/*v* agarose gel electrophoresis.

### 2.6. Species Identification

DNA extracted from individual mosquitoes was used for molecular species identification. The mitochondrial genes COI and COII were sequenced in morphologically identified *An. pharoensis* samples, according to Krzywinski et al. [15], after cleaning-up the PCR products. PCR amplification reactions were carried out in 20 µL volume reaction mix, containing 1xPCR buffer, 250 µM of each DNTP, 2.5 mM MgCl_2_, 0.15 mg/mL of bovine serum albumin, one unit of Kappa Taq polymerase, 1 µL of genomic DNA and 600 nM of each primer. PCR product size ranged from 700 bp to 730 bp. For the amplification of COI the following primers were used F (5′-3′): GGA GGATTTGGAAATTGATTAGTTCC; R (5′-3′): GCTAATCATCTAAAAATTTTAATTCC; whereas for COII the following set of primers were used F (5′-3′): TCTAATATGGGAGATTAGTGC; R (5′-3′): ACTTGCTTTCAGTCATCTAAT G. The PCR conditions were 3 min at 95 °C followed by 30 s at 95 °C, 30 s at 55 °C and 45 s at 72 °C for 35 cycles and 10 min at 72 °C for the final extension. The PCR products were then separated by electrophoresis on 1.5% agarose gel with Midori green and visualized under ultraviolet light.

### 2.7. kdr Detection in Anopheles Mosquitoes

Direct sequencing (Sanger) of the VGSC gene, including the area of the L1014F/S mutation, was applied to *An. pharoensis* samples to detect the presence of *kdr* mutations. Initial PCR analyses were conducted to select the area of interest for sequencing. PCR amplification reactions were carried out in 20 µL volume reaction mix, containing 1xPCR buffer, 250 µM of each DNTP, 2 mM MgCl_2_, 0.15 mg/mL of bovine serum albumin, one unit of Kappa Taq polymerase, 1 µL of genomic DNA and 500 nM of each of following primers. For the amplification of *kdr* the following were used as primers F (5′-3′): GGMGAATGGATYGAATCMATGTGGGA; R (5′-3′): GATGAACCRAAATTKGACAAAAGCAA. The PCR conditions were 5 min at 94 °C followed by 1 min at 94 °C, 2 min at 50 °C and 2 min at 72 °C for 35 cycles and 2 min at 72 °C for the final extension. The PCR products were then separated by electrophoresis on 1.5% agarose gel with Midori green and visualized under ultraviolet light.

### 2.8. Cuticular Hydrocarbons (CHCs) Identification and Quantitation by GC-MS and GC-FID

Female mosquitoes from the sites of Yangah (Y), Simatou (S) and Maga (M) were processed. Before analysis, mosquitoes were air-dried at 25 °C. Then, air-dried mosquitoes were pooled (25 female mosquitoes/population, three biological replicates), their dry weight was measured and the corresponding samples proceeded for CHC analysis. CHC identification and quantitation (by GC-MS and GC-FID) was performed as previously described in [16], with minor modifications. Briefly, cuticular lipids from all fifteen samples were extracted by 1 min immersion in hexane (×3) with gentle agitation; extracts were pooled and evaporated under a N_2_ stream. CHCs were separated from other components and finally concentrated prior to chromatography by Solid Phase Extraction (SPE). Quantitative amounts were estimated by co-injection of nC24 as an internal standard (2890 ng/mL in Hexane). CHC quantitation was calculated as the sum of area of 32 peaks in total, using the internal standard.

## 3. Results

### 3.1. Species Identification

A total of 30 specimens (10 per site) were processed in order to confirm the morphological identification using Sanger sequencing of mitochondrial COI/COII genes, and all mosquitoes were confirmed as *An. pharoensis* (Table 1).

### 3.2. Bioassay Analysis

A total of 201 *An. pharoensis* were exposed to DDT 4% to determine their susceptibility profile. All the three populations appeared to be resistant to DDT 4%, with mortality rate ranging from 62.79% in Maga to 80% in Yangah (Table 2).

### 3.3. kdr L1014F/S Analysis in An. pharoensis Samples

Out of the 114 *An. pharoensis* mosquitoes expressing phenotypic resistance sequenced, none were detected carrying the *kdr* allele L1014F (*kdr* W) and/or L1014S (*kdr* E) (Table 3 and Figure 1). All mosquito analyzed turned out to be of the wild type.

### 3.4. Analysis of Amount of Cuticular Hydrocarbon (CHC) in Mosquito

In the present study, the mean amount of CHCs normalized for dry body weight (ng CHCs/mg dry body weight ± SD) from the three *An. pharoensis* mosquito populations (Yangah (Y), Simatou (S) and Maga (M)) was calculated as 2420.9 ± 265, 2372.5 ± 225, 1409.7 ± 105 ng CHCs/mg, respectively (Figure 2). The difference of the means between Yangah and Maga and Simatou and Maga populations was found to be significant (two-tailed *t*-test, *p* = 0.0036 **, *p* = 0.0026 **, respectively). The raw data of the analysis are presented in Appendix A.

## 4. Discussion

The study objective was to assess the profile of DDT resistance in *Anopheles pharoensis*. High resistance to DDT was recorded in all study sites. The low level of susceptibility detected in *An. pharoensis* populations was consistent with the general profile of DDT resistance in *An. gambiae* and *An. funestus* populations across Cameroon. characterized by no or very low mortality rates to this compound [17,18,19]. To our knowledge, this is the first report of insecticide resistance in *An. pharoensis* in Cameroon. The present result confirms the emergence of insecticide resistance in other mosquito species. Recent studies in Cameroon also reported increase tolerance of species such as *An. nili, An. paludis* and *An. rufipes* to pyrethroids [6,20]. *An. pharoensis* has been reported resistant to DDT in Ethiopia using standard WHO insecticide bioassay [11]. The emergence of resistance in *An. pharoensis* could result from the selective pressure induced by the use of LLINs or pesticides in agriculture. This species breeds mainly in permanent and semi-permanent habitats such as irrigation canals, rice field swamps and lake shores which could accumulate a large number of xenobiotics which could exert a high selective pressure on mosquitoes breeding in such habitats during a longer period. It is also possible that resistance could spread through gene flow between close *An. pharoensis* populations. Molecular identification of species using the COI and COII confirm the presence of *An. pharoensis* in all study sites. However, it is not clear whether *An. pharoensis* populations distributed across Africa are genetically homogeneous or differentiated due to the existence of factors that could restrict gene flow between populations, such as geographical distance or physical barriers (mountains), or due to the patchy distribution of breeding habitats. Further investigations using new available genetic tools are needed. Studies conducted so far in Cameroon indicated that *An. pharoensis* feeds equally indoors and outdoors [14], suggesting that the species’ susceptibility status could also be affected by the use of insecticide-treated nets. However, molecular analysis suggested no implication of *kdr* mutation in *An. pharoensis* resistance to DDT.

Based on the study from Balabanidou et al. [16], which showed that pyrethroid-resistant *An. gambiae* mosquitoes had increased epicuticle thickness, mainly because of a substantially higher amount of CHCs compared to susceptible mosquitoes, we could suggest that Yangah and Simatou field populations could have a thicker epicuticle compared to the Maga population, and this could therefore help uptake insecticide molecules slower. In their study, Talipouo et al. [21] reported a CHC content of 1840 ± 70 ng CHCs/mg dry weight for the resistant samples, whereas the corresponding value was 1552 ± 80.1 ng CHCs/mg dry weight for the susceptible strain. Our data support the implication of cuticular resistance in *An. pharoensis* resistance to DDT in the sites of Yangah and Simatou. Two mechanisms are considered to reduce the penetration of insecticide in mosquitoes; these include: the cuticle thickening and the altering of cuticle composition [22]. These processes could be under the actions of genes or proteins such as *Cyp4g16*, laccase 2 or ABC transporters [22]. It is likely that additional mechanisms, such as metabolic detoxification, are involved in DDT resistance in *An. pharoensis* populations, but this warrants further investigation. Several mechanisms, including target sites and metabolic-based mechanisms, have been reported to be involved in resistance to insecticides in mosquitoes [23].

## 5. Conclusions

The present study highlights the need for the intensification of surveillance activities on different vector species contributing to malaria transmission across Africa. *An. pharoensis,* as other secondary malaria vector, is still largely overlooked by control programmes. In order to achieve malaria elimination, more consideration needs to be given to all neglected vectors which contribute to the maintenance of malaria transmission.

## Figures and Tables

**Figure 1 genes-13-01723-f001:**
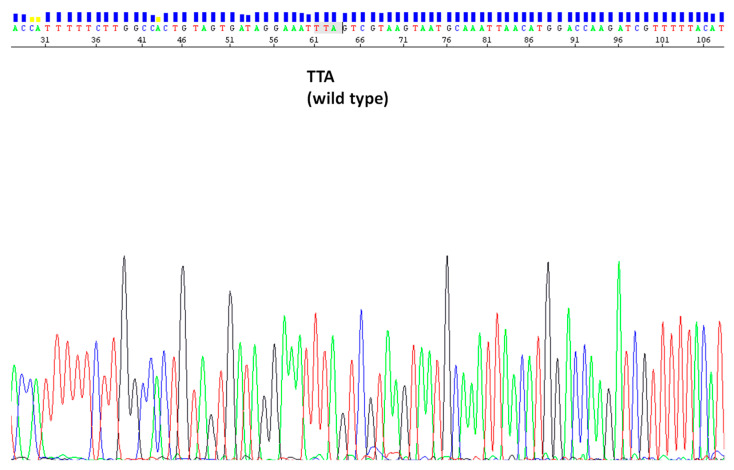
Representative electropherogram of a wild-type (TTA) *An. pharoensis* sample for the *kdr* L1014F mutation.

**Figure 2 genes-13-01723-f002:**
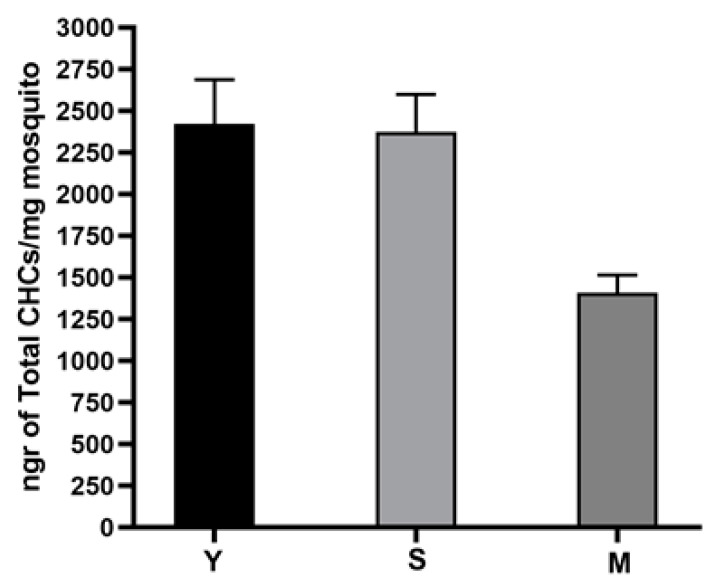
Mean cuticular hydrocarbon (CHC) amounts from three Anopheles mosquito populations. Yangah (Y) and Simatou (S) *An. pharoensis* have significantly higher amounts of CHCs compared to Maga (M) mosquitoes normalized for their size differences (two-tailed *t*-test, *p* value < 0.05).

**Table 1 genes-13-01723-t001:** Molecular identification of *An. pharoensis* from the Yangah, Simatou and Maga populations after amplification of COI and COII genes.

Population	N	Species ID
Yangah	10	100% *An. pharoensis*
Simatou	10	100% *An. pharoensis*
Maga	10	100% *An. pharoensis*

N: Number of mosquitoes.

**Table 2 genes-13-01723-t002:** Susceptibility status of *An. pharoensis* populations to 4% DDT.

Sites/Strain	Species	Tested	Dead	Mortality Rate (95% CI)
Maga	*An. pharoensis*	86	54	62.79% (47.2–81.9)
Simatou	*An. pharoensis*	90	69	76.67% (59.65–97.03)
Yangah	*An. pharoensis*	25	20	80% (48.87–123.55)
Kisumu	*An. gambiae*	100	100	100%

95% CI: confidence interval.

**Table 3 genes-13-01723-t003:** Genotyping of *An. pharoensis* mosquitoes for the L1014F/S mutation.

Population	Sample Size (Alleles)	Phenotype	*kdr* L1014F/S Genotype
Yangah	20	DDT resistant	Wild type (TTA or TTG)
Simatou	20	DDT resistant	Wild type (TTA or TTG)
Maga	74	DDT resistant	Wild type (TTA or TTG)
Kisumu	30	susceptible strain	Wild type (TTA or TTG)

## Data Availability

All the data from the study are included in the manuscript.

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
