# Peer review of "DDT Resistance in Anopheles pharoensis from Northern Cameroon Associated with High Cuticular Hydrocarbon Production"

_genes, 2022, doi:10.3390/genes13101723_

Round 1

Reviewer 1 Report

I have the following suggestions/corrections to the authors

1. Under section 2.3, bioassays, It is appropriate to mention that mosquitoes were transferred to holding tubes and fed for 24 h before scoring the mortalities. This clearly describes what you did especially to readers who know little about insect testing 

2. under section 2.6

You carried out species identification, it is not clear if you did this using the DNA extracted from individual mosquitoes or the pooled samples you described somewhere. Be specific and clear on these. 

3. You have sequenced the VGSC of a good sample size, I am wondering why you did not carry out further sequence analysis to possibly see if there are any allele selections despite the lack of kdr mutations. Probably you have other important mutations somewhere in the VGSC? 

Author Response

Reviewer 1 comments

Comment 1:  

Under section 2.3, bioassays, it is appropriate to mention that mosquitoes were transferred to holding tubes and fed for 24 h before scoring the mortalities. This clearly describes what you did especially to readers who know little about insect testing. 

Response 1:

We thank the reviewer for the comment the sentence was changed accordingly see section on “Insecticide bioassay”,

Comment 2:

Under section 2.6

You carried out species identification, it is not clear if you did this using the DNA extracted from individual mosquitoes or the pooled samples you described somewhere. Be specific and clear on these. 

Response 2:

DNA extracted from individual mosquito was used for molecular identification of species. This precision has been provided in the manuscript.

Comment 3

You have sequenced the VGSC of a good sample size, I am wondering why you did not carry out further sequence analysis to possibly see if there are any allele selections despite the lack of kdr mutations. Probably you have other important mutations somewhere in the VGSC? 

Response 3

We thank the reviewer for this suggestion we will consider these analyses in our future studies.

Reviewer 2 Report

I sincerely apologize for the inconvenience. I have attached my comments and recommendations in picture form due to the current internet outage in my area. 

Author Response

Reviewer 2 comments

Comment 1

Be consistence when it comes to species name Anopheles pharoensis

Response 1

Thanks your remark was taken into consideration in the manuscript

Comment 2

Table 1. be more specific on what the numbers within the parenthesis represent? How many times did the author perform this experiment?

Response 2

The numbers within the parenthesis represent 95% confidence interval. Information added on the footnote of table 2.

Comment 3

Why Table 2 is before Table 1? Also where is Table 3?

Response 3

Sorry it’s an error this was corrected.

Comment 4

Table 4. I recommend the author produce the sequencing data for the genotyping analysis to further support their claim.

Response 4

Electropherogram of a wild-type (TTA) An. pharoensis sample for the kdr L1014F mutation was added in figure 1.

Comment 5

What is ngr on the y-axis?Maybe”Total CHCs/mg from pooled mosquitoes collected from their study sites. I suggest the authors should include another control.

Response 5 

ngr CHCs/mg mosquito means nanogram of total cuticular hydrocarbons per mg mosquito weight. Control unfortunately cannot be added because mosquitoes were analysed randomly and comparisions were done between them. 
CHC is a trait that is affected by enviromental conditions (growth 
conditions, laboratory conditions etc). Comparisons of strains that grow similtaneously or natural populations are valid.